# Humanized single domain antibodies neutralize SARS-CoV-2 by targeting the spike receptor binding domain

Xiaojing Chi[1,4], Xiuying Liu[1,4], Conghui Wang[2,4], Xinhui Zhang[1], Xiang Li[3], Jianhua Hou[3], Lili Ren[2], Qi Jin[1✉], Jianwei Wang [2✉] & Wei Yang [1✉]

Severe acute respiratory syndrome coronavirus 2 (SARS-CoV-2) spreads worldwide and leads to an unprecedented medical burden and lives lost. Neutralizing antibodies provide efficient blockade for viral infection and are a promising category of biological therapies. Here, using SARS-CoV-2 spike receptor-binding domain (RBD) as a bait, we generate a panel of humanized single domain antibodies (sdAbs) from a synthetic library. These sdAbs reveal binding kinetics with the equilibrium dissociation constant ($K_D$) of 0.99–35.5 nM. The monomeric sdAbs show half maximal neutralization concentration ($EC_{50}$) of 0.0009–0.07 μg/mL and 0.13–0.51 μg/mL against SARS-CoV-2 pseudotypes, and authentic SARS-CoV-2, respectively. Competitive ligand-binding experiments suggest that the sdAbs either completely block or significantly inhibit the association between SARS-CoV-2 RBD and viral entry receptor ACE2. Fusion of the human IgG1 Fc to sdAbs improve their neutralization activity by up to ten times. These results support neutralizing sdAbs as a potential alternative for antiviral therapies.

[1]NHC Key Laboratory of Systems Biology of Pathogens, Institute of Pathogen Biology, Chinese Academy of Medical Sciences and Peking Union Medical College, Beijing, China. [2]NHC Key Laboratory of Systems Biology of Pathogens and Christophe Mérieux Laboratory, Institute of Pathogen Biology, Chinese Academy of Medical Sciences and Peking Union Medical College, Beijing, China. [3]Beijing Kawin Technology Co., Ltd., Beijing, China. [4]These authors contributed equally: Xiaojing Chi, Xiuying Liu, Conghui Wang. ✉email: zdsys@vip.sina.com; wangjw28@163.com; wyang@ipb.pumc.edu.cn

Coronavirus disease 2019 (COVID-19) is caused by infection of emerging severe acute respiratory syndrome-associated coronavirus 2 (SARS-CoV-2) and had been declared by World Health Organization as the first coronavirus pandemic in human history[1]. The severity of COVID-19 symptoms can range from asymptomatic or mild to severe with an estimated mortality rate from less than 2% to up to 10% of patients depending on various factors[2]. SARS-CoV-2 is spreading rapidly and sustainably around the world, urging prompt global actions to develop vaccines and antiviral therapeutics.

SARS-CoV-2 polyprotein shares ~86.15% identity with SARS-CoV (Genbank ID: AAS00002.1) and is classified into the genus betacoronavirus in the family *Coronaviridae*[3]. SARS-CoV-2 is an enveloped, positive-sense, single-stranded RNA virus with a large genome of approximately 30,000 nucleotides in length. The virus-encoded membrane (M), spike (S), and envelope (E) proteins constitute the majority of the protein that is incorporated into SARS-CoV-2 envelope lipid bilayer. The S protein can form homotrimers and protrudes from envelope to show the coronal appearance, invading susceptible cells by binding potential SARS-CoV-2 entry receptor angiotensin converting enzyme 2 (ACE2)[3]. Recently, researchers have figured out the molecular structure of SARS-CoV-2 S protein[4]. It is composed of 1273 amino acids and structurally belongs to the type I membrane fusion protein with two areas S1 and S2. The S1 region mainly includes the receptor binding domain (RBD), while the S2 region is necessary for membrane fusion. The RBD structure determines its binding efficiency with ACE2 and provides an important target for neutralizing antibody recognition.

Single domain antibodies (sdAbs), namely nanobodies, were initially identified from camelids or cartilaginous fish heavy-chain only antibodies devoid of light chains, where antigen-binding is mediated exclusively by a single variable domain (VHH)[5]. Therefore, sdAbs are the smallest fragments that retain the full antigen-binding capacity of the antibody with advantageous properties as drugs, imaging probes and diagnostic reagents[6]. The advantages of short development time, flexible formatting and robust production efficiency make sdAb a powerful means to defeat infectious disease pandemics. For therapeutic purpose, relatively sophisticated humanization techniques have been adopted to modify the camelid-specific amino acid sequences in the framework to their human heavy chain variable domain equivalent, without altering sdAb's biological and physical properties and reducing species heterogeneity[7]. As SARS-CoV-2 is an emerging human virus, the whole population is susceptible due to the lack of protective antibodies. The existing neutralizing antibodies in convalescent plasma have been adopted as powerful therapeutic alternatives for COVID-19 patients.

In this study, using a synthetic humanized sdAbs discovery platform, we obtain several high-affinity SARS-CoV-2 RBD targeting sdAbs with desired neutralization activities. The results illustrate the potential of synthetic sdAb library as a resource for antiviral molecules that can be rapidly accessed in a pandemic. These sdAbs offer a potential hope for future anti-SARS-CoV-2 antibody cocktails.

## Results

**Identification of SARS-CoV-2 RBD binding sdAbs.** SARS-CoV-2 makes use its envelope S glycoprotein to gain entry into host cells through binding ACE2. Recent cryo-EM research revealed that the S protein shows an asymmetrical homotrimer with a single RBD in the "up" confirmation and the other two "down"[4]. Antibodies may take advantage of this RBD structure to block virus entry. To enrich for SARS-CoV-2 RBD binding sdAbs, we performed four rounds of biopanning using a lab owned, full

synthetic, humanized phage display library with recombinant RBD protein. After phage ELISA identification of 480 clones, a number of sdAbs exhibited an excellent affinity for SARS-CoV-2 RBD (Supplementary Table 1). Five distinctive sdAd sequences (1E2, 2F2, 3F11, 4D8, and 5F8) were cloned into a prokaryotic expression vector and recombinant sdAb proteins were purified by nickel-charged sepharose affinity chromatography (Fig. 1a). Humanized sdAbs obtained in this study are about 125 amino acids with a single VHH domain in average molecular weight less than 15KDa (Fig. 1a). The sdAbs consist of three complementarity determining regions (CDRs), as well as four framework regions (FRs). The amino acids in the frameworks have been maximally humanized, except for residues Phe-42 and Ala-52 (numbers refer to the International ImMunoGeneTics information system amino acid numbering (imgt.cines.fr)) in framework-2 to maintain proper antigen affinity and best stability[7]. Framework residues are illustrated in Supplementary Fig. 1.

Surface plasmon resonance (SPR) technology is widely accepted as a golden standard for characterizing antibody-antigen interactions. To determine the kinetic rate and affinity constants, detailed analysis of Spike RBD-binding to purified sdAb proteins was carried out by SPR. The SARS-CoV-2 or SARC-CoV RBD protein was immobilized on the surface of Biacore Chip CM5, respectively. Then, various concentrations of purified sdAbs were prepared and injected to pass over the surface. The sensorgram data were fitted to a 1:1 steady-state binding model. SPR results demonstrated that the equilibrium dissociation constant ($K_D$) for the SARS-CoV-2 RBD protein against sdAbs 1E2, 2F2, 3F11, 4D8, and 5F8 were 35.52 nM, 5.175 nM, 3.349 nM, 6.028 nM, and 0.996 nM, respectively (Fig. 1b–f, h). However, the sdAbs showed no binding with SARS-CoV RBD, except for the clone 5F8 demonstrating a relatively low affinity with $K_D = 239.2$ nM (Fig. 1g, h). Overall, as monovalent antibody fragment, the sdAbs identified in this study reveals a satisfactory binding performance in a SARS-CoV-2 specific manner.

**Neutralization of SARS-CoV-2 by RBD-specific sdAbs.** To further evaluate the neutralization activity of these sdAbs, SARS-CoV-2 Spike-pseudotyped particle (SARS-CoV-2pp) infectivity assay was first established. Pseudotyped particles are chimeric virions that consist of a surrogate viral core with a heterologous viral envelope protein at their surface, which can be operated in Biosafety Level 2 (BSL-2) and frequently used tool for studying virus entry mechanism and neutralizing antibodies[8]. We observed that all five sdAbs showed inhibition potency of SARS-CoV-2pp infection with $EC_{50}$ (half maximal neutralization concentration) ranging from 0.0009 to 0.069 μg/mL (Fig. 2a). We next tested the neutralization activity of the sdAbs with SARS-CoV-2 live virus (Fig. 2b). The copy number of viral RNA that was present in the cell culture supernatant was used as a proxy for viral replication. Similarly, these sdAbs showed comparable neutralization efficiency, with $EC_{50}$ at approximately 0.13–0.51 μg/mL. Totally, these monovalent sdAbs demonstrated encouraging neutralization activity against both pseudotyped and authentic virus, although the neutralization potency is not completely matched (Fig. 2c). This phenomenon was normally reported in Middle East Respiratory Syndrome coronavirus (MERS-CoV) neutralizing antibodies and may be likely explained by the difference in sdAb recognized RBD spatial epitope or the steric hindrance formed by antigen-antibody complex[9,10].

**Interference of the ACE2-RBD interaction by the sdAbs.** Within SARS-CoV-2 RBD, the receptor binding motif (RBM) directly contacts ACE2. Recent report demonstrating that SARS-

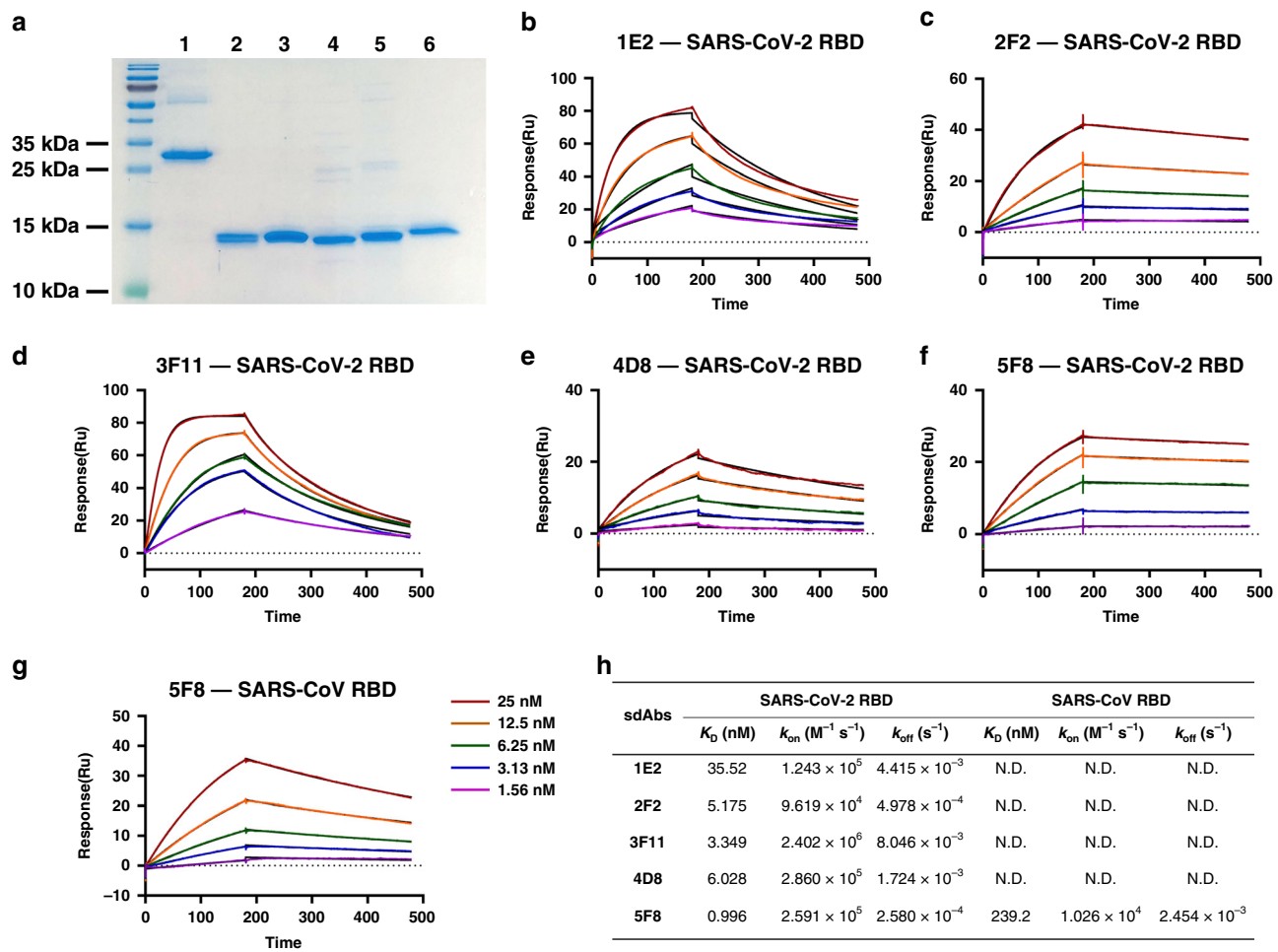

**Fig. 1 Identification of SARS-CoV-2 RBD binding sdAbs. a** The purified recombinant proteins of SARS-CoV-2 RBD and sdAbs were separated by SDS-PAGE and stained with Coomassie Blue. Lanes: 1, SARS-CoV-2 RBD; 2, 1E2; 3, 2F2; 4, 3F11; 5, 4D8; 6, 5F8. **b–f** Five sdAbs binding to SARS-CoV-2 RBD measured by SPR. Two-fold serial dilutions from 25 nM sdAb injected onto the captured RBD protein. Kinetic data from one representative experiment were fit to a 1:1 binding model. The profiles are shown for 1E2 (**b**), 2F2 (**c**), 3F11 (**d**), 4D8 (**e**), and 5F8 (**f**). **g** Kinetics of binding between SARS-CoV RBD and sdAb 5F8. **h** Summary of SPR kinetic and affinity measurements. The equilibrium dissociation constant ($K_D$), the association constant ($k_{on}$) and the dissociation constant ($k_{off}$) are presented. N.D. means not detected.

CoV-2 uses ACE2 as its receptor with a much stronger affinity (10-fold to 20-fold higher) than SARS-CoV[4]. To determine whether sdAbs targeted different antigenic regions on the SARS-CoV-2 RBD surface, we performed a competition-binding assay using a real-time biosensor (Fig. 3). We tested all five sdAbs in a competition-binding assay in which human ACE2 was attached to a CM5 biosensor. Compared with a non-related isotype control sdAb (Fig. 3a), addition of 1E2 and 4D8 completely prevent binding of SARS-CoV-2 RBD to ACE2 (Fig. 3b, e). Whereas, sdAbs 2F2, 3F11, and 5F8 could partially compete the RBD/receptor association (Fig. 3c, d, f). These data suggested that these sdAbs can be divided into RBM targeting or non-RBM targeting groups though it is not directly associated with either affinity or virus neutralization activity, which has laid a solid foundation for further development of bispecific neutralizing antibodies to overcome potential virus mutation in the future.

**Inhibition of SARS-CoV-2 entry by Fc-fused sdAbs.** SdAbs can be readily fused to human IgG Fc-domain to overcome the limitations of the monovalent sdAbs, such as the short blood residential time and lacking antibody-dependent cell-mediated cytotoxicity and complement dependent cytotoxicity[11]. In addition, bivalent sdAbs can be obtained via the disulfide bond formation in Fc hinge area, which was reported to increase sdAb's activity[12]. To further explore the possibility of sdAb-based antiviral therapeutics and enhance neutralization activity, we constructed human heavy chain antibodies by fusing the human IgG1 Fc region to the C-terminus of sdAbs (Fig. 4a, b). These Fc fusion sdAbs were produced in mammalian cells with supernatant yields around 25–50 μg per milliliter in shaking flask. Fc fusion sdAbs in culture supernatants were affinity purified with HiTrap Protein A HP antibody purification columns (Supplementary Fig. 2) and analyzed in both reducing and non-reducing conditions in Western blot using an anti-human IgG to detect Fc. As shown in Fig. 4c, the size of the constructed intact sdAb-Fc is around 80 KDa in the non-reducing condition, but a 40 KDa monomer was observed by prior treatment in reducing condition to break disulfide bonds. This suggests a correct expression and secretion of heavy chain antibodies in consistence with our design. Neutralization assay results showed that genetic fusion of human Fc could maintain or increase the neutralization activity of these sdAbs for up to 10-fold in molar concentration of $EC_{50}$ using the SARS-CoV-2pp entry assay (Fig. 4d and Supplementary Fig. 3). Importantly, all Fc-fused sdAbs demonstrated potency with $EC_{50}$ at sub-nanomolar level (Fig. 4d). Finally, we showed that some of

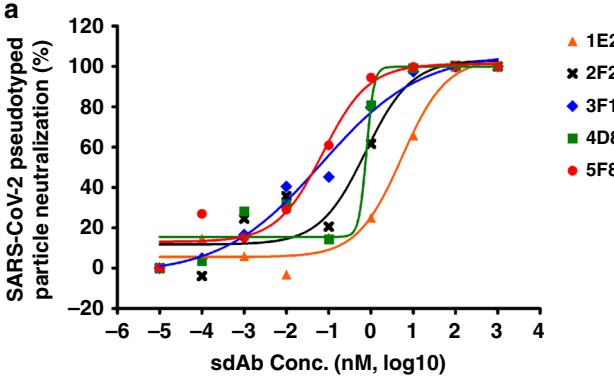

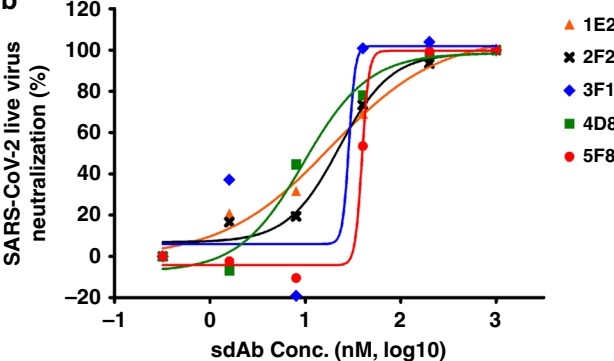

**c**

| sdAbs | EC$_{50}$, nM (μg/mL) | |
|---|---|---|
| | pseudotyped particle | live virus |
| 1E2 | 5.324 (0.0692) | 18.47 (0.2401) |
| 2F2 | 0.742 (0.0096) | 22.62 (0.2941) |
| 3F11 | 0.066 (0.0009) | 28.64 (0.3723) |
| 4D8 | 0.781 (0.0102) | 9.628 (0.1252) |
| 5F8 | 0.072 (0.0009) | 39.28 (0.5107) |

**Fig. 2 Neutralization of SARS-CoV-2 by RBD-specific sdAbs.**
**a** Neutralization of 5 sdAbs against SARS-CoV-2pp. SARS-CoV-2pp was pre-incubated with 5-fold serially diluted sdAbs before inoculation of human ACE2 transfected 293T cells. At 48 h post infection, luciferase activities were measured, and percent neutralization was calculated. The experiments were performed independently at least twice and similar results were obtained. One representative data of one experiment were shown and data were average values of three replicates ($n = 3$).
**b** Determination of neutralization activities of 5 sdAbs against live SARS-CoV-2. Absolute quantification of SARS-CoV-2 RNA copy number in culture supernatants was performed using real time RT-PCR method, and percent neutralization was calculated. The experiments were performed independently at least twice and similar results were obtained. One representative data of one experiment were shown and data were average values of two replicates ($n = 2$). **c** Summary of the half maximal neutralization concentration (EC$_{50}$) values of the 5 sdAbs against both SARS-CoV-2pp and live virus.

the sdAbs are suitable for immunofluorescence staining (Supplementary Fig. 4) and Western blot to detect ectopically expressed SARS-CoV-2 S protein (Supplementary Fig. 5).

## Discussion

Given the disease severity and rapid global spread of COVID-19, there is an urgent need for development of vaccines, monoclonal antibodies, and small-molecule direct-acting antiviral medications. Neutralizing antibodies directly target viral envelope protein,

precisely block the virus-receptor association, and inhibit virus entry through a variety of molecular mechanisms. In this study, we isolated and characterized several humanized neutralizing sdAbs that exhibit one-digit to two-digit nanomolar or even sub-nanomolar EC$_{50}$ against SARS-CoV-2 using both pseudotyped and infectious viruses. SdAbs have been investigated as important therapeutic alternatives against viral infection because of their high yield, low cost and intrinsic stability. For MERS-CoV, neutralizing sdAbs were isolated from immunized dromedary camels or llamas and demonstrated EC$_{50}$ value between 0.001 and 0.003 μg/mL with low $K_D$ values (0.1–1 nM)[13,14]. Comparable inhibition efficiency on SARS-CoV-2pp and affinity kinetics were obtained for the sdAbs identified in this study using a non-immune library, which can speed up the discovery of neutralizing antibodies in an emergent outbreak. With further optimization and increase of library size and diversity, the synthetic sdAb library technology will promote the discovery speed of powerful therapeutic antibodies[15,16].

FDA approved the first sdAb-based medicine for adults with acquired thrombotic thrombocytopenic purpura in 2019[17–20]. Considering the cost and potential risks of full human antibody in some viral diseases, such as dengue virus infection, sdAb fragments are a novel category of therapeutic molecules and can be readily reconstructed in a tandemly linked way to increase their blood residential time, biological activity, and eliminate underlying concerns about antibody-dependent enhancement (ADE) of viral infection[21]. In addition to being used as an injectable drug, the stable sdAbs can be also developed into aerosolized inhalations and disinfection products for the prevention of COVID-19. Besides, prior to the success of COVID-19 vaccines, the construction of sdAb-based adenovirus or adeno-associated virus gene therapy might provide long-term passive immune protection in vulnerable population, health care workers, or in severely affected areas. Since the mature COVID-19 animal models have not been developed, this study did not involve in vivo studies. As a next step, the crystal structure analysis of antigen-antibody complexes will be put on the agenda. In conclusion, the discovered neutralizing antibodies in this study could lead to new specific antiviral treatments and shed light on the design and optimization of COVID-19 vaccines.

## Methods

**Cells and reagents**. The Vero (African green monkey kidney), HEK293T (human kidney epithelial), 293F cells were obtained from China Infrastructure of Cell Line Resource (Beijing, China) and maintained in Dulbecco's modified Eagle's medium (DMEM, ThermoFisher, Waltham, MA, USA) supplemented with 2–10% fetal bovine serum (FBS, ThermoFisher), non-essential amino acid, penicillin and streptomycin. Recombinant proteins were purchased from Sino Biological (Beijing, China) for SARS-CoV-2 RBD (40592-V05H, 40592-V08B), SARS-CoV RBD (40150-V08B2), ACE2 (10108-H08H) and the recombinant Fc region of mouse IgG1 (10690-MNAH). Antibodies were obtained from ThermoFisher for anti-His-HRP (MA1-21315-HRP), anti-human IgG-HRP (31410), anti-His-488 (MA1-21315-A488). HRP/anti-CM13 monoclonal conjugate was from GE Healthcare (27-9421-01, 1:3000).

**Library design and construction**. A synthetic sdAb phage display library was used for the screening of SARS-CoV-2 neutralizing antibodies. To minimize a possible antigenic effect from camelid sequences, sdAb frameworks (FRs) for library construction were determined according to a universal humanized scaffold architecture[7], and the sequences of the FRs were illustrated in Supplementary Fig. 1. Briefly, residues in FRs 1, 3, and 4 were mutated based on human heavy chain VH in maximum. In FR 2, humanization of residues at positions 49 and 50 was adopted to increase stability of sdAbs, whereas residues 42 and 52 are maintained in camelid due to their critical impact on antigen affinity and/or stability (Supplementary Fig. 1). For the design of variable regions, we analyzed a robust CDR repertoire from immune or naïve llama VHH clones. A synthetic diversity was introduced in the three CDRs by the positioned nucleotide assembly with cysteine and stop codon avoided. A constant length of 8 amino acids was selected for CDR1 and CDR2, and 18 amino acids for CDR3 (Supplementary Fig. 1). Frameworks and CDRs were assembled using only 8 cycles of overlapping polymerase chain reaction (PCR) to prevent drift during amplification. Diversified sdAb mixture was cloned

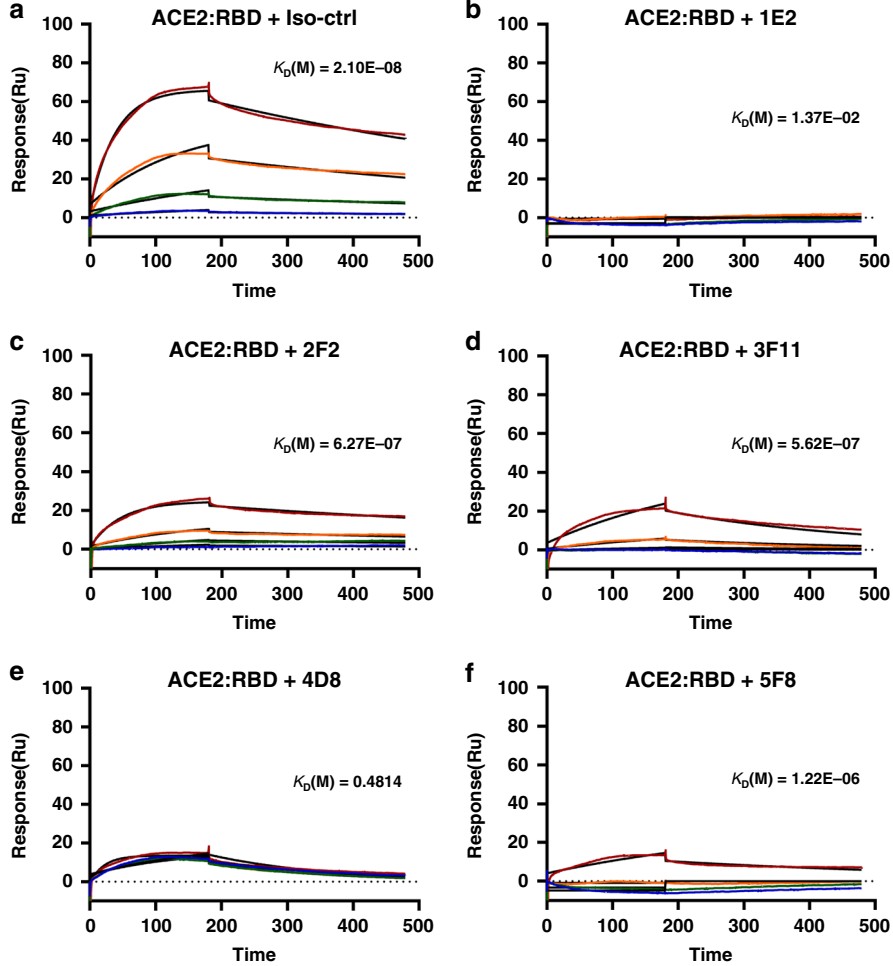

**Fig. 3 Interference of the ACE2-RBD interaction by the sdAbs.** The recombinant human ACE2 protein was immobilized on CM5 chip using a BIAcore T200 machine and tested for the binding with gradient concentrations of SARS-CoV-2 RBD that were diluted in 100 nM sdAbs, including an isotype control sdAb (**a**), 1E2 (**b**), 2F2 (**c**), 3F11 (**d**), 4D8 (**e**), and 5F8 (**f**).

in phagemid vector fADL-1 (Antibody Design Labs, San Diego, CA, USA) using SfiI/BglI sites with the PelB peptide leader sequence fused with the sdAbs at N-terminus. The ligation product was purified and used to transform electro-competent E. coli TG1 cells. A total 50 electroporations was performed in the condition of 1800 V, 10 mF, 600 W. Each electroporation was resuspended with 2 ×YT and incubated with a shaking agitation for 1 h at 37 °C, and then combined and plated onto more than a thousand agar petri dishes (140 mm) to ensure enough size of the library. Library size was calculated by plating serial dilution aliquots and at least $1.2 \times 10^{10}$ individual recombinant clones were obtained. Quality control was carried out by sequencing more than 1000 clones. More than 950 clones are full length and unique sdAbs and less than 50 clones show various errors, such as vector self-ligation, reading frame shift and fragment deletion.

**Antibody selection by phage display.** Screening for SARS-CoV-2 RBD targeting antibodies was performed by panning in both immunotubes and native condition using a proprietary full-synthetic library of humanized sdAbs with high-diversity, according to a standard protocol. Briefly, for the 2nd and 4th panning rounds, the purified SARS-CoV-2 RBD protein fused with mouse Fc was coated on Nunc MaxiSorp immuno tubes (ThermoFisher) at around 5 μg/mL in PBS overnight. For the 1st and 3rd panning rounds, RBD protein was first biotinylated with EZ-Link™ Sulfo-NHS-LC-Biotin (ThermoFisher) and then selected with streptavidin-coated magnetic Dynabeads™ M-280 (ThermoFisher). The tubes or beads were blocked using 2% w/v skimmed milk powder in PBS (MPBS). After rinsing with PBS, about $1 \times 10^{13}$ phage particles were added to the antigen-coated immuno tube or bioti-nylated antigen in the presence of 2% MPBS, incubated for 2 h shaking (30 rpm) at RT. Unbound phages were washed with PBS Tween 0.1% (10 times) and PBS (10 times), while bound phage were eluted with 0.1 M Glycine-HCl (pH = 3.0). Eluted phages were neutralized by adding 1 M Tris-Cl pH 9.0 and used for infection of exponentially growing E. coli TG1. After 4 rounds of panning, phage ELISA identification was performed with 480 individual colonies using Anti-CM13 anti-body [B62-FE2] (HRP) in the plates coated with either mouse Fc-fused SARS-CoV-2 RBD or mouse Fc as a negative control for screening. The absorbance was

measured using a SpectraMax M5 plate reader from Molecular Divices (San Jose, CA, USA). The positive clones were determined according to the criteria of SARS-CoV-2 RBD positive and mouse Fc negative and sent for sequencing. After sequence alignments, 5 distinctive sdAb sequences were chosen for protein expression.

**Expression and purification of sdAbs.** Full-length sequences of selected sdAbs were PCR amplified and cloned into the NcoI/XhoI sites of the pET28b (Novagen, Sacramento, CA, USA) and transformed into BL21(DE3) chemically competent E. coli. A single colony was picked to inoculate 10 ml of LB media containing Kanamycin (100 μg/mL) and incubated at 37 °C on an orbital shaker overnight. This preculture was diluted 1:100 in 400 mL of LB media containing Kanamycin (100 μg/mL) and grown at 37 °C until the OD$_{600}$ nm reached 0.4. The expression of recombinant sdAbs was induced by adding IPTG to a final concentration of 0.3 mM after culture has reached OD$_{600}$ = 0.5–0.6 and grown over night at 20 °C. The sdAbs with a His-tag fused at C-terminus were purified over Ni Sepharose 6 Fast Flow (GE Healthcare, Boston, MA, USA) and eluted with 400 mM imidazole. Affinity purified sdAbs were dialyzed against PBS to eliminate imidazole.

**Affinity measurement and competition-binding study.** The surface plasmon resonance experiments were performed at room temperature using a BiaCore T200 with CM5 sensor chips (GE Healthcare). The surfaces of the sample and reference flow cells were activated with a 1:1 mixture of 0.1 M NHS (N-hydroxysuccinimide) and 0.1 M EDC (3-(N,N-dimethylamino) propyl-N-ethylcarbodiimide) at a flow rate of 10 μL/min. The reference flow cell was left blank. All the surfaces were blocked with 1 M ethanolamine, pH 8.0. The running buffer was HBS-EP (0.01 M HEPES, pH 7.4, 150 mM NaCl, 3 mM EDTA, 0.05% surfactant P20).

For binding affinity assays, the His-tagged SARS-CoV-2 RBD or SARS-CoV RBD was diluted in 10 mM sodium acetate buffer, pH5.5, and was immobilized on the chip at about 300 response units. Antibodies 1E2, 2F2, 3F11, 4D8, and 5F8 at gradient concentrations (0, 1.56 nM, 3.125 nM, 6.25 nM, 12.5 nM, 25 nM) were

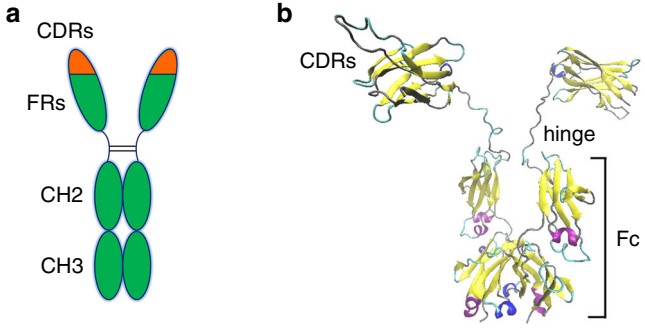

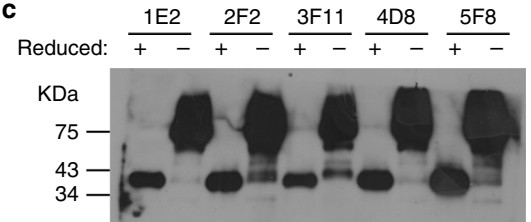

| sdAb-Fc | EC$_{50}$<br>nM (μg/mL) | EC$_{50}$<br>fold increase (nM) |
|---------|-----------------|----------------------------|
| 1E2-Fc | 0.538 (0.043) | 9.9 |
| 2F2-Fc | 0.400 (0.032) | 1.9 |
| 3F11-Fc | 0.013 (0.001) | 5.1 |
| 4D8-Fc | 0.463 (0.037) | 1.7 |
| 5F8-Fc | 0.055 (0.004) | 1.3 |

**Fig. 4 Inhibition of SARS-CoV-2 entry by Fc-fused sdAbs. a** Representation of the human IgG1 Fc-fused sdAbs in this study. SdAb-Fc fusion construction generates a bivalent molecule with an approximate molecular weight of 80 kDa. **b** Homology modeling of the bivalent 5F8-Fc molecule with SWISS-MODEL server (https://swissmodel.expasy.org)[23]. The template structure for 5F8 modeling was based on a humanized camelid sdAb in the PDB database (3EAK). The structure is depicted as cartoons and colored with secondary structure. Three CDRs, hinge region and Fc were indicated. **c** Five Fc-fused sdAbs were analyzed by Western blot with gradient SDS-PAGE in reducing (with β-mercaptoethanol) or non-reducing (without β-mercaptoethanol) condition. **d** Summary of EC$_{50}$ value of Fc-fused sdAbs neutralization against SARS-CoV-2pp. EC$_{50}$ fold increases versus the corresponding monovalent sdAbs were calculated.

flowed over the chip surface. After each cycle, the sensor surface was regenerated with10 mM glycine-HCl pH 2.5. The data were fitted to a 1:1 interaction steady-state binding model using the BIAevaluation 1.0 software.

For competition-binding assays, the ACE2 protein was diluted in 10 mM sodium acetate buffer, pH4.5, and was immobilized on the chip at about 650 response units. For the analyses, the His-tagged SARS-CoV-2 RBD protein was diluted in HBS-EP buffer or HBS-EP buffer with 100 nM antibody (1E2, 2F2, 3F11, 4D8, or 5F8). The RBD in different buffer at gradient concentrations (0, 6.25 nM, 25 nM, 100 nM and, 400 nM) was flowed over the chip surface. After each cycle, the sensor surface was regenerated with 10 mM glycine-HCl pH 2.5. The binding kinetics was analyzed with the software of BIAevaluation using a 1:1 binding model.

**SARS-CoV-2 spike pseudotyped particle (SARS-CoV-2pp).** To produce SARS-CoV-2pp, HEK293T cells were seeded 1 day prior to transfection at $2.5 \times 10^6$ cells in a 10-cm plate. The next day, cells were transfected using Lipofectamine 2000 (ThermoFisher). The plasmid DNA transfection mixture (1 ml) was composed of 15 μg of pNL-4.3-Luc-E$^-$R$^-$ and 15 μg of pcDNA-SARS-CoV-2-S that was purchased from Sino Biologicals and reconstructed by deletion of 18 amino acid cytoplasmic tail. A nonenveloped lentivirus particle (Bald virus) was also generated as negative control. 16 h after transfection, the media was replaced with fresh media supplemented with 2% FBS. Supernatants containing SARS-CoV-2pp were typically harvested at 36–48 h after transfection and then filtered through a syringe filter (0.22 μm) to remove any cell debris. SARS-CoV-2pp was freshly used or allocated and frozen at −80 °C. To conduct the virus entry assay, 293T cells were transiently transfected with human ACE2 expression plasmid and $1 \times 10^4$ cells and seeded in each well of a 96-well plate at 1 day prior to transduction. The next day,

100 μL of supernatant containing SARS-CoV-2pp was added into each well in the absence or presence of serially diluted sdAbs or human IgG1 Fc-fused sdAbs. Forty-eight hours after transduction, the cells were lysed in 100 μL of passive lysis buffer and 50 μL lysate was incubated with 100 μL of luciferase assay substrate according to the manufacturer's instructions (Promega, Madison, WI, USA).

**Ethics statement and virus isolation.** SARS-CoV-2 was isolated from bronchoalveolar lavage fluid (BALF) from a COVID-19 patient in the Jin Yin-tan Hospital of Wuhan as reported previously[22]. Briefly, the patient was a 65-year-old man who reported a high fever and cough, with little sputum production, at the onset of illness. He had a continuous fever and developed severe shortness of breath 16 days later. BALF sample was collected from this hospitalized patient by nurses according to a standard procedure in which a bronchoscope is passed through the mouth into the lungs to obtain cells and other components from bronchial and alveolar spaces. A clinical protocol was conducted in accordance with the Declaration of Helsinki and was approved by the National Health Commission of China and Ethics Commission of the Jin Yin-tan Hospital of Wuhan (No. KY-2020-01.01). Written informed consent was waived by the Ethics Commission of the designated hospital for emerging infectious diseases. Clearing the airway and collection of BALF were as standard of care and for clinical etiological diagnosis. Therefore, the requirement for written informed consent was waived given the context of emerging infectious diseases. For the isolation and identification of potential pathogens, the BALF specimens were filtered and inoculated onto Vero cells. All cultures were observed daily for a cytopathic effect (CPE). CPE were observed in 30% of Vero cells after two passages. The viral particles in culture supernatants were characterized by negative staining electron microscope. The isolated SARS-CoV-2 was obtained from the patient by Dr. Lili Ren and the virus full length sequence was deposited in GISAID database with accession ID of EPI_ISL_402123, which is completely as same as GenBank accession number MN908947. GISAID is a globally recognized virus database and more than 56,000 viral genomic sequences of hCoV-19 have been shared via GISAID since the start of the COVID-19 outbreak.

**SARS-CoV-2 neutralization assay.** The 50% tissue culture infectious dose (TCID$_{50}$) assay was performed for SARS-CoV-2 in Vero cells. Briefly, cells were seeded 24 h before infection in a 24-well plate at a density of $8 \times 10^4$ cells/well. Viruses were serially diluted at 1:10 dilution. After 72 h of incubation, the media were removed, and cells were fixed and stained with crystal violet. The TCID$_{50}$/ml titer was determined. For antibody neutralization assay, Vero cells were seeded in 96-well plates at 1 day prior to infection. Serially diluted sdAbs were mixed with SARS-CoV-2 at 100 TCID$_{50}$ per well and incubated at 37 °C for 1 h. The antibody-virus mixture was incubated on Vero cells at 37 °C for 1 h. Unbound SARS-CoV-2 virions were removed by washing cells with fresh medium, then incubated for 24 h at 37 °C. The culture supernatants were collected for viral nucleic acid quantification. Viral RNA quantification was carried out by TaqMan real-time RT-PCR as reported with plotted standard curves using in vitro transcribed RNA. Briefly, the viral RNA was isolated using TRIzol LS reagent (Invitrogen, Carlsbad, CA) according to the manufacturer's protocol. RNA was extracted from 100 μL culture supernatants and eluted in 50 μL DNase/RNase-Free water. The Viral nucleocapsid gene-based quantification assay was developed using the TaqMan Fast Virus 1-Step Master Mix (Applied Biosystems, Foster City, CA) on CFX96™ Real-Time PCR System (Bio-Rad, Hercules, CA). Each 20 μL reaction mixture contained 5 μL of 4× Fast virus 1-step Master Mix, 1 μL of RNA, primers (2019-nCoV_N1-F: 5′-GAC CCC AAA ATC AGC GAA AT-3′ and 2019-nCoV_N1-R: 5′-TCT GGT TAC TGC CAG TTG AAT CTG-3′) at working concentration of 20 μM and 5 μM probe (5′-FAM-ACC CCG CAT TAC GTT TGG TGG ACC-BHQ1-3′). The standard curve is composed by 5 standards with serial dilutions of in vitro transcribed and quantified RNAs ($10^3$, $10^4$, $10^5$, $10^6$, and $10^7$). All amplifications were performed using the CFX96™ Real-Time PCR System.

**Production of human IgG1 Fc fusion sdAbs.** The sequences of selected sdAbs were cloned into a mammalian expression vector under the control of hEF1-HTLV promotor and fused with N-terminal interleukin-2 signal peptide and C-terminal Fc region, comprising the CH2 and CH3 domains of human IgG1 heavy chain and the hinge region. Maxiprepped plasmids were transiently transfected into 293-F cells (Thermofisher) and the cells were further cultured in suspension for 6 days before harvesting antibody-containing supernatant. Fc-fused sdAbs were prepared with prepacked HiTrap® Protein A HP column (GE Healthcare). The produced Fc-fusion protein was analyzed by SDS-PAGE and the Western blot using standard protocols for dimerization, yield and purity measurement. The primary antibody used for Western blot was a horseradish peroxidase conjugated goat anti-human IgG (Sigma-Aldrich, St. Louis, MO, USA).

**Immunofluorescence microscopy and Western blot.** Cultured 293T cells on coverslips were transfected with either SARS-CoV-2 S expression plasmid or empty vector for 24 h and then fixed using 4% paraformaldehyde for 15 min at room temperature, permeabilized with 0.1% Triton X-100 (Sigma-Aldrich) in PBS for 10 min. The cells were then incubated with each sdAb overnight at 4 °C. After three washes with PBS, the cells were incubated with Alexa Fluor 488–conjugated 6x-His Tag monoclonal antibody (HIS.H8) (ThermoFisher, MA1-21315-A488, 1:1000) for

1 h at room temperature. The nuclei were stained with DAPI (1:10,000) diluted in PBS for 5 min and mounted with an antifade reagent (ThermoFisher). Images were acquired with a Leica TCS SP5 confocal microscope system.

For Western blot, 293T cells in 6-well plate were transfected with SARS-CoV-2 S, SARS-CoV-2 S or empty vector individually. Twenty-four h post transfection, cell lysates were prepared, and the samples were boiled with 2× SDS loading buffer and loaded onto a 10% polyacrylamide gel. After electrophoresis, the separated proteins were transferred onto a nitrocellulose membrane (Bio-Rad, Hercules, CA, USA). The resulting blots were probed with a sdAb as primary antibody and an HRP-linked 6x-His Tag antibody (Thermofisher, HIS.H8, MA1-21315-HRP, 1:1000) as the secondary antibody. Antibody against β-Actin is from Sigma-Aldrich (A1978, 1:8000). The ECL reagent (Amersham Biosciences, Piscataway, NJ, USA) was used as the substrate for detection.

**Statistics and reproducibility**. Data were analyzed using GraphPad Prism 6.01 (GraphPad Software, San Diego, CA, USA). The values shown in the graphs are presented as means ± SD. One representative result from at least two independent experiments was shown. Antibody neutralization experiments usually use two to four duplicated wells for each treatment. For SARS-CoV-2pp entry assay and SARS-CoV-2 infection, the infectivity data were first inversed to neutralization activity. Each neutralization data set was normalized by the background control (no virus) to define the real value for 100% neutralization. After transformation to neutralization, the lowest concentration point of antibody treatment was set to 0% neutralization. Then, a 4-parameters neutralization nonlinear regression model was fitted to report $EC_{50}$ values. All experiments were performed independently at least twice and similar results were obtained. One representative data of one experiment were shown.

**Reporting summary**. Further information on research design is available in the Nature Research Reporting Summary linked to this article.

## Data availability
The sequences of sdAbs have been deposited in GenBank with the accession numbers MT813428-MT813432. The isolated SARS-CoV-2 full length sequence was deposited in GISAID database with accession ID of EPI_ISL_402123, which is completely as same as GenBank accession number MN908947. All other data are available from the corresponding author upon reasonable requests. Source data are provided with this paper.

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

## Acknowledgements
This work was supported by CAMS Initiative for Innovative Medicine Grant 2020-I2M-2-010 and 2016-I2M-3-020.

## Author contributions
W.Y., X.C., L.R. Q.J., and J.W. designed experiments and interpreted the data. W.Y., X.C., X.L., C.W., X.L., J.H., and X.Z. performed experiments and analyzed the data. W.Y. conceived the study, supervised the work, and wrote the paper. All authors read and approved the final manuscript.

## Competing interests
A patent application has been filed on 17 March 2020 on single domain antibodies targeting SARS-CoV-2 (China patent application no. 202010185593.9).
