## [Peer Review File · Nature Communications]

REVIEWER COMMENTS

Reviewer #1 (Remarks to the Author):

The authors report on 5 single domain antibodies that can neutralize SARS-CoV-2 S pseudotyped viruses as well as SARS-CoV-2. These VHs originate from a synthetic camelid sdAb phagemid library in which frame work regions were partially humanized and nucleotide diversity was introduced in CDR1 (length 8 aas), CDR2 (length 8 aas) and CDR3 (length 18 aas). The 5 sdAbs were selected after 4 rounds of panning on immobilized SARS-CoV-2 RBD followed by phage ELISA. The sdAbs bind with variable affinity to SARS-CoV-2 RBD or S and can neutralize lentivirus pseudotypes with an estimated IC50 value that ranges from 3 to 300 ng/ml. In contrast, all 5 sdAbs neutralized live SARS-CoV-2 with a similar IC50 (0.24-0.51 ug/ml). The 2 sdAbs that could completely prevent the binding of RBD to immobilized ACE2 performed poorest in the pseudotype neutralization assay. The sdAbs were also genetically fused to a human IgG1 Fc domain, expressed in 293-F cells and purified, and shown to be able to neutralizes pseudotyped lentiviruses, with an IC50 down to 1.6 ng/ml for 2F2-Fc.

The work is of interest because it illustrates the potential of synthetic sdAb libraries as a resource for antiviral molecules that can be rapidly accessed, e.g. in case of a pandemic. A major caveat of the manuscript is the poor description of the synthetic library, the neutralization assays and the experiments in general. The sequences of the 5 sdAbs is not shown and should be provided.

Major remarks:

1. The synthetic sdAb library must be better described. What are the sequences of the FRs? The primers used to introduce diversity in the CDRs? What is the complexity of the library. On page 10 the authors write "More than a thousand agar perti dishes..." How many colonies were obtained in total? "Quality control was carried out by sequencing more than 1000 clones, and the error rate and diversity was calculated". That is essential information that should be shared, because it documents the diversity of the library, uniqueness of individual clones, lack of in frame stop codons, percentage of phagemids without or with a truncated insert.
2. The sequences of the 5 selected sdAbs as well as the irrelevant control sdAb should be provided.
3. The panning and selection of the 5 sdAbs must be explained better. Panning was performed with RBD (biotinylated or not). However, RBD fused to a mouse Fc or RBD with a His-tag are both mentioned (line 215). How was sdAb binding to Fc avoided? After 4 rounds of panning 480 individual colonies were obtained and tested in phage ELISA, from which "representative sdAb sequences were chosen for protein expression". What was the exact set up of the phage ELISA and what was the criterion to select those 5 sdAbs?
4. Figure 1 and page 5. Line 112 states and affinity of 0.676 nM of 5F8 for S, whereas Figure 1F depicts 0.715. Please clarify. Figure 1 and its legend do not accord: in the figure RBD is mentioned throughout whereas the legend and lines 110-114 mention "S". Please clarify.

5. The pseudotype particle generation should be explained in more detail in the material and methods section: “filtered through a syringe filter”? “Vero cells or other cells seeded ..” Which other cells, cell density?
6. It is interesting that 3F11 and 5F8, which only partially suppress ACE2-RBD interaction can not completely neutralize the pseudotype particles. The authors should provide an explanation for this finding.
7. The SARS-CoV-2 neutralization assay has to be better described (line 315). How was the virus isolated from the BAL of the patients? Propagated on which cells? How many passages? How many vero cells were seeded? The amount of viral RNA that was present in the cell supernatant was used as a proxy for viral replication. Again essential protocol details are missing. How was RNA extracted, cDNA synthesized, which primer and probes were used, how was standard RNA generated and quantified? Figure 2b, Y-axis: percentage of what?
8. Figure 4. A Coomassie stained gel should be included to document the purity of the preparations. Line 337: why was a goat anti-mouse IgG antibody used to stain the human Fc fusions in the Western blot?
9. Supplementary figure 2 is not visible.

Reviewer #2 (Remarks to the Author):

Xiaojing Chi and colleagues report the isolation and characterization of nanobodies targeting the receptor-binding domain of SARS-CoV-2. This is a very timely and important study but major revisions are needed before it could be published.

Major comments:

The authors describe the selection of RBD binders using a phage display, but they do not provide any data showing the selection/improvement process during the cycles they performed. This data should be included, even if it goes to the supplementary information.

It is not clear what the authors mean by saying that they “maximally humanized” the framework of the binders. Nanobodies are inherently non-human, and it is not clear how “human-like” they become after this design. The authors will need to provide additional data. Also, the authors

mention several times that some residues were mutated, while others were kept as appear in camelids to maintain stability. A major question relates to the actual stability of the designs, as low stability will make them irrelevant for further development. It will be important to evaluate their thermal stabilities and compare them to non-modified nanobodies.

The neutralization/infectivity data that are shown in Figure 2 are poorly presented, and perhaps also inadequately treated. The authors present infectivity data, but there are huge variations in the infectivity levels in low reagent concentrations. A dashed line is shown at “50%”, but since some of the reagents plateau at ~60%, this line is meaningless. The authors do not specify how IC50 values were calculated, and thus the validity of these values is questionable. To make sense, the data should be inversed to show “neutralization” rather than “infectivity”. Also, each neutralization data set should be normalized by the “background” (no virus) control to define the real value for 100% neutralization. Then, a 4-parameters neutralization model should be fitted and shown with the experimental data points. When fitting this model, it will be acceptable to force it to reach 0% neutralization at the lowest concentration of the reagents. From this model, the IC50, as well as IC90 values, should be reported. There is a good chance that the reported IC50 values will change following this treatment.

For all the SPR data shown in the manuscript, the authors need to present along with the experimental data (overlaid) the fitted model from which the kinetic parameters were derived. For clarity, it will be convenient to use a two-color scheme. Residual plots (in the supplementary) will also be desired. Lastly, there is a convention for how to present chemical quantities like using italics for the ‘K’ and subscript for the ‘D’ in the “KD”. The same goes true for the on and off rate parameters.

The authors present the production of Fc-fusion constructs in figure 4 and they report the IC50 and IC80 values for these constructs. It is imperative that the authors will also present the neutralization data (could be in the supplementary data), treated as mentioned above.

Response to referees

NCOMMS-20-12472

Revision: Humanized Single Domain Antibodies Neutralize SARS-CoV-2 by Targeting Spike Receptor Binding Domain. By Chi *et al.*

Reviewer #1:

General remarks: The authors report on 5 single domain antibodies that can neutralize SARS-CoV-2 S pseudotyped viruses as well as SARS-CoV-2. These VHHs originate from a synthetic camelid sdAb phagemid library in which frame work regions were partially humanized and nucleotide diversity was introduced in CDR1 (length 8 aas), CDR2 (length 8 aas) and CDR3 (length 18 aas). The 5 sdAbs were selected after 4 rounds of panning on immobilized SARS-CoV-2 RBD followed by phage ELISA. The sdAbs bind with variable affinity to SARS-CoV-2 RBD or S and can neutralize lentivirus pseudotypes with an estimated IC₅₀ value that ranges from 3 to 300 ng/ml. In contrast, all 5 sdAbs neutralized live SARS-CoV-2 with a similar IC₅₀ (0.24-0.51 ug/ml). The 2 sdAbs that could completely prevent the binding of RBD to immobilized ACE2 performed poorest in the pseudotype neutralization assay. The sdAbs were also genetically fused to a human IgG1 Fc domain, expressed in 293-F cells and purified, and shown to be able to neutralizes pseudotyped lentiviruses, with an IC₅₀ down to 1.6 ng/ml for 2F2-Fc. The work is of interest because it illustrates the potential of synthetic sdAb libraries as a resource for antiviral molecules that can be rapidly accessed, e.g. in case of a pandemic. A major caveat of the manuscript is the poor description of the synthetic library, the neutralization assays and the experiments in general. The sequences of the 5 sdAbs is not shown and should be provided.

General response: *Thank you for your detailed and very helpful comments, all of them have been responded to thoroughly and included in the revision presently submitted. Please find our point-by-point responses as follows.*

Specific remarks:

Q #1: “The synthetic sdAb library must be better described. What are the sequences of the FRs? The primers used to introduce diversity in the CDRs? What is the complexity of the library. On page 10 the authors write “More than a thousand agar petri dishes...” How many colonies were obtained in total? “Quality control was carried out by sequencing more than 1000 clones, and the error rate and diversity was calculated”. That is essential information that should be shared, because it documents the diversity of the library, uniqueness of individual clones, lack of in frame stop codons, percentage of phagemids without or with a truncated insert.”

Our response: *According to this suggestion, we have added a more detailed description in the Materials and Methods section, including but not limited to library construction and quality validation, live virus neutralization, as well as statistical determination of neutralization activity. For the synthetic sdAb library, detailed methodologies are provided. In addition, the design and sequence of sdAb scaffold are summarized in Supplementary Fig. 1. Regarding the antibody screening process, we supplemented the ELISA identification of the clones obtained from the fourth round of screening, as shown in Supplemental Table 1.*

Q #2: “The sequences of the 5 selected sdAbs as well as the irrelevant control sdAb should be provided.”

Our response: *We understand the requirement regarding the disclosure of antibody sequences. However, the antibody sequence is in a pending patent application. In order to contribute to the global effort to combat the novel coronavirus pandemic, we would like to commit to providing the purified antibodies and necessary sequence information to non-profit organizations and individuals through the signing of an MTA.*

Q #3: “The panning and selection of the 5 sdAbs must be explained better. Panning

was performed with RBD (biotinylated or not). However, RBD fused to a mouse Fc or RBD with a His-tag are both mentioned. How was sdAb binding to Fc avoided? After 4 rounds of panning 480 individual colonies were obtained and tested in phage ELISA, from which “representative sdAb sequences were chosen for protein expression”. What was the exact set up of the phage ELISA and what was the criterion to select those 5 sdAbs?”

Our response: *We appreciate these suggestions and supplemented some new descriptions in Method section accordingly. Actually, in the beginning, only a mouse Fc-fused recombinant SARS-CoV-2 RBD was commercially available, so it was used for library screening. Discovery of RBD specific sdAb clones was guaranteed by using recombinant mouse Fc as a negative screen control during ELISA in parallel, as shown in Supplemental Table 1. Subsequently, we were able to purchase a His-tagged SARS-CoV-2 RBD protein that was applied in SPR experiments for affinity analysis. The process for the selection of representative sdAb sequences is as follows: sequencing was performed for SARS-CoV-2 RBD positive but mouse Fc negative clones; sequence alignment for excluding repeated clones; determination for distinctive clones for protein expression.*

Q #4: “Figure 1 and page 5. Line 112 states and affinity of 0.676 nM of 5F8 for S, whereas Figure 1F depicts 0.715. Please clarify. Figure 1 and its legend do not accord: in the figure RBD is mentioned throughout whereas the legend and lines 110-114 mention “S”. Please clarify.”

Our response: *Thank you for pointing out these inconsistencies. 0.676 nM is correct, and correction was made in Fig. If accordingly. In addition, all the proteins used in this study were recombinant RBD, not Spike. Corrections have been made throughout the paper.*

Q #5: “The pseudotype particle generation should be explained in more detail in the

material and methods section: “filtered through a syringe filter”? “Vero cells or other cells seeded ..” Which other cells, cell density?”

Our response: *We appreciate reviewer for this constructive suggestion. More detailed descriptions were added for pseudoparticle production and live virus neutralization assay in Materials and Method section.*

Q #6: “It is interesting that 3F11 and 5F8, which only partially suppress ACE2-RBD interaction can not completely neutralize the pseudotype particles. The authors should provide an explanation for this finding.”

Our response: *Thank you for this observation. If you look at the experimental points, the high antibody concentrations can completely neutralize the infection. It may be due to the problem of fitting the curve, which seems unable to completely neutralize the pseudotyped virus. In this revised version, we transformed the results from an infection-inhibition effect to a more recognized neutralization effect, replotted Figure 2 and Figure 4. The results in the revised Figure 2 show that all 5 single domain antibodies can completely neutralize the virus.*

Q #7: “The SARS-CoV-2 neutralization assay has to be better described (line 315). How was the virus isolated from the BAL of the patients? Propagated on which cells? How many passages? How many vero cells were seeded? The amount of viral RNA that was present in the cell supernatant was used as a proxy for viral replication. Again essential protocol details are missing. How was RNA extracted, cDNA synthesized, which primer and probes were used, how was standard RNA generated and quantified? Figure 2b, Y-axis: percentage of what?”

Our response: *We appreciate your suggestions very much. Substantial revision has been made for this part.*

Q #8: “Figure 4. A Coomassie stained gel should be included to document the purity

of the preparations. Line 337: why was a goat anti-mouse IgG antibody used to stain the human Fc fusions in the Western blot?”

Our response: *A Coomassie stained gel was demonstrated in Supplementary Fig. 2 to show the purity and correct dimerization. Thank you for pointing this mistake. It is anti-human IgG antibody in the Western blot. Correction has been made.*

Q #9: “Supplementary figure 2 is not visible.”

Our response: *This may be caused by the website conversion of PDF files, we have adjusted it.*

Reviewer #2:

General remarks: Xiaojing Chi and colleagues report the isolation and characterization of nanobodies targeting the receptor-binding domain of SARS-CoV-2. This is a very timely and important study but major revisions are needed before it could be published.

Specific comments:

Q #1: “The authors describe the selection of RBD binders using a phage display, but they do not provide any data showing the selection/improvement process during the cycles they performed. This data should be included, even if it goes to the supplementary information.”

Our response: *We appreciate reviewer for this constructive suggestion. First, a more detailed description about library construction and screening was added in Materials and Methods section. Second, discovery of SARS-CoV-2 RBD specific sdAb clones via 4 rounds biopanning was guaranteed by using recombinant mouse Fc as a negative screen control during ELISA in parallel, as shown in Supplemental Table 1. The*

process for the selection of representative sdAb sequences is as follows: sequencing was performed for SARS-CoV-2 RBD positive but mouse Fc negative clones; sequence alignment for excluding repeated clones; determination for distinctive clones for protein expression.

Q #2: “It is not clear what the authors mean by saying that they “maximally humanized” the framework of the binders. Nanobodies are inherently non-human, and it is not clear how “human-like” they become after this design. The authors will need to provide additional data. Also, the authors mention several times that some residues were mutated, while others were kept as appear in camelids to maintain stability. A major question relates to the actual stability of the designs, as low stability will make them irrelevant for further development. It will be important to evaluate their thermal stabilities and compare them to non-modified nanobodies.”

Our response: *We agree with you for this concern. To make this clear, detailed methodologies are provided. In addition, the design and sequence of humanized sdAb scaffold are summarized in Supplementary Fig. 1. Actually, the library construction strategy and scaffold humanization are completely based on a published work. Vincke et al. General strategy to humanize a camelid single-domain antibody and identification of a universal humanized nanobody scaffold. J Biol Chem. 2009; 284(5): 3273-84. In their paper, considerable work was carried out to investigate the relationship between scaffold humanization and antibody expression yield, antigen affinity, biochemical Properties and thermodynamic stability. Conclusively, this universal humanization method provides an excellent strategy to construct a single-framework, synthetic library by introducing variability into its CDRs. Our work is formally based on the findings from this published reference.*

Q #3: “The neutralization/infectivity data that are shown in Figure 2 are poorly presented, and perhaps also inadequately treated. The authors present infectivity data,

but there are huge variations in the infectivity levels in low reagent concentrations. A dashed line is shown at “50%”, but since some of the reagents plateau at ~60%, this line is meaningless. The authors do not specify how IC₅₀ values were calculated, and thus the validity of these values is questionable. To make sense, the data should be inverted to show “neutralization” rather than “infectivity”. Also, each neutralization data set should be normalized by the “background” (no virus) control to define the real value for 100% neutralization. Then, a 4-parameters neutralization model should be fitted and shown with the experimental data points. When fitting this model, it will be acceptable to force it to reach 0% neutralization at the lowest concentration of the reagents. From this model, the IC₅₀, as well as IC₉₀ values, should be reported. There is a good chance that the reported IC₅₀ values will change following this treatment.”

Our response: *Thank you very much for your advice. We have conducted new statistical analysis and data processing according to this recommendation, and the relevant results have been transformed from the inhibition efficiency of the virus to the neutralization efficiency of the virus. Also, IC₅₀ was transformed to EC₅₀ though the values changed slightly after this treatment. In this revised version, you will find Figure 2 and Figure 4 have been updated. At the same time, we also give a more detailed description of the Materials and Methods section.*

Q #4: “For all the SPR data shown in the manuscript, the authors need to present along with the experimental data (overlaid) the fitted model from which the kinetic parameters were derived. For clarity, it will be convenient to use a two-color scheme. Residual plots (in the supplementary) will also be desired. Lastly, there is a convention for how to present chemical quantities like using italics for the ‘K’ and subscript for the ‘D’ in the “KD”. The same goes true for the on and off rate parameters.”

Our response: *We appreciate your suggestions very much. Required revision was made to demonstrate the original experimental data and the fitted model by using black and color scheme overlaid (Figure 1b-1g). In addition, the affinity representation has also*

been normalized. Thanks again for your advice.

Q #5: “The authors present the production of Fc-fusion constructs in figure 4 and they report the IC50 and IC80 values for these constructs. It is imperative that the authors will also present the neutralization data (could be in the supplementary data), treated as mentioned above.”

Our response: *These results were revised accordingly, and the viral neutralization curves of the Fc-fusion antibodies were also shown in Supplementary Fig. 3.*

REVIEWER COMMENTS

Reviewer #1 (Remarks to the Author):

The authors have made reasonable efforts to improve the manuscript.

The library is still not very well described. "Sequence analysis of more than 1000 clones, and the error rate was less than 5%". It is important to provide the number of different clones that were represented in this over 1000 clones sample. Were all sequenced clones full length and with a unique sequence?

The clinical sample is not well described. The sequence of the virus was likely determined, which the authors should deposit.

The selection of the 5 clones based on the phage ELISA seems to be more or less random: why were e.g. wells 1.1G, 2.1B, 2.1F, 2.7E, 4.9F, 4.5B, 5.9D, 5.4E and 5.3G not selected for further analysis?

It is unclear why the authors avoid to publish the sequences of their SDABs, especially since IP has been filed already.

Reviewer #2 (Remarks to the Author):

The authors addressed most of my comments but some additional revisions are needed.

Outstanding issues:

1) The SPR data and analysis in Figure 1, should be improved. While for some nanobodies the data and model look OK (i.e. 5F8), for others like 1E2, 2F2, and 5F8 the fitted models do not adequately describe the data. The models were fitted with a refractive index (RI) change when it is clear that the data do not show that. Better models should be provided, perhaps even two-state models if a simple

1:1 binding model cannot be fitted to the data. The authors will need to adjust their reported affinity values accordingly.

2) Another issue is the reported values in Figures 1h, 2c, 4d, and throughout the text. It looks like too many significant figures are used to describe some of the values.

3) Figure 3 is lacking a binding model that should be shown overlaid on the experimental data when a KD is reported. Also, "KD" is not written properly.

4) The ELISA data in table S1 is meaningless to most readers. This data should be presented in a graphical way. Please make separate graphs for each round and perhaps one that demonstrates the round to round improvements.

Response to referees

NCOMMS-20-12472A

Revision: Humanized Single Domain Antibodies Neutralize SARS-CoV-2 by Targeting Spike Receptor Binding Domain. By Chi *et al.*

Reviewer #1:

General remarks: The authors have made reasonable efforts to improve the manuscript.

Specific remarks:

Q #1: “The library is still not very well described. "Sequence analysis of more than 1000 clones, and the error rate was less than 5% ". It is important to provide the number of different clones that were represented in this over 1000 clones sample. Were all sequenced clones full length and with a unique sequence?”

Our response: *We actually picked 1000 clones randomly for sequencing. More than 950 clones are full length and unique sdAbs sequences, and less than 50 clones show various errors, such as vector self-ligation, reading frame shift or fragment deletion. Therefore, we described that the error rate was less than 5% in the previous version. Revision was made in Method section accordingly in page 10.*

Q #2: “The clinical sample is not well described. The sequence of the virus was likely determined, which the authors should deposit.”

Our response: *2. The methodology used for virus isolation, source and clinical sampling was detailed described in this revised version in page 13. Briefly, the patient was a 65-year-old man who reported a high fever and cough, with little sputum production, at the onset of illness. He had a continuous fever and developed severe shortness of breath 16 days later. BALF sample was collected from this hospitalized*

patient by clinical doctors according to a standard procedure in which a bronchoscope is passed through the mouth into the lungs to obtain cells and other components from bronchial and alveolar spaces.

The isolated SARS-CoV-2 full length sequence was deposited in GISAID with Accession ID of EPI_ISL_402123 (<https://platform.gisaid.org/epi3/frontend#57c81d>). GISAID is a globally recognized virus database and more than 56,000 viral genomic sequences of hCoV-19 have been shared via GISAID since the start of the COVID-19 outbreak. I have registered in GISAID. Editors and reviewers are allowed to login GISAID and browse our deposited sequence with the following information (user ID: yangwei, password: qmxVHUKG). In addition, the metadata and FASTA sequence were attached in the end of this point-by-point response to the reviewers.

Q #3: “The selection of the 5 clones based on the phage ELISA seems to be more or less random: why were e.g. wells 1.1G, 2.1B, 2.1F, 2.7E, 4.9F, 4.5B, 5.9D, 5.4E and 5.3G not selected for further analysis?”

Our response: *After phage ELISA, we sequenced all the positive clones (including 1.1G, 2.1B, 2.1F, 2.7E, 4.9F, 4.5B, 5.9D, 5.4E and 5.3G) and analyzed their sequences. Most of the positive clones are repeated sequences. Finally, five unique and representative clones with neutralizing activities were selected for further study. They are 1E2, 2F2, 3F11, 4D8 and 5F8.*

Q #4: “It is unclear why the authors avoid to publish the sequences of their SDABs, especially since IP has been filed already.”

Our response: *we understand the requirements of the disclosure of antibody sequences. We are committed to providing antibody sequences and associated reagents to researchers around the world by signing the MTA because, during submission and revision, our antibodies were licensed to a pharmaceutical company for development and optimization, and we cannot disclose the antibody sequence according to the*

licensing agreement. We are sorry for the inconvenience. We hope that our commitment to providing antibody sequence to nonprofit academic societies will meet your requirement.

Reviewer #2:

General remarks: The authors addressed most of my comments but some additional revisions are needed.

Specific comments:

Q #1: “The SPR data and analysis in Figure 1, should be improved. While for some nanobodies the data and model look OK (i.e. 5F8), for others like 1E2, 2F2, and 5F8 the fitted models do not adequately describe the data. The models were fitted with a refractive index (RI) change when it is clear that the data do not show that. Better models should be provided, perhaps even two-state models if a simple 1:1 binding model cannot be fitted to the data. The authors will need to adjust their reported affinity values accordingly.”

Our response: *We agree with you for this concern. The fitted model has been redone. The models were still fitted to a 1:1 interaction steady-state binding model. We used blanks from other concentration series, which list at the bottom of the table for blank subtraction. The affinity has been adjusted accordingly.*

Q #2: “Another issue is the reported values in Figures 1h, 2c, 4d, and throughout the text. It looks like too many significant figures are used to describe some of the values.”

Our response: *We understand this concern about so many values in the figures. The purpose of this presentation is to give readers a more intuitive view of the important parameters and indicators in the development of these antibodies.*

Q #3: “Figure 3 is lacking a binding model that should be shown overlaid on the experimental data when a KD is reported. Also, “KD” is not written properly.”

Our response: *Thank you very much for your advice. We have revised Figure 3 accordingly.*

Q #4: “The ELISA data in table S1 is meaningless to most readers. This data should be presented in a graphical way. Please make separate graphs for each round and perhaps one that demonstrates the round to round improvements.”

Our response: *Thanks for this comment. As you may know, we usually do not analyze the enrichment and improvement in each round of screening. We only pick clones for phage ELISA after the last round biopanning.*

Virus detail	
Virus name:	hCoV-19/Wuhan/IPBCAMS-WH-01/2019
Accession ID:	EPI_ISL_402123
Type:	betacoronavirus
Lineage (GISAID Clade):	B (L)
Passage details/history:	Original
Sample information	
Collection date:	2019-12-24
Location:	Asia / China / Hubei / Wuhan
Host:	Human
Additional location information:	
Gender:	Male
Patient age:	65
Patient status:	
Specimen source:	Bronchoalveolar lavage fluid
Additional host information:	
Outbreak:	
Last vaccinated:	
Treatment:	
Sequencing technology:	
Assembly method:	
Coverage:	
Comment:	
Institute information	
Originating lab:	Institute of Pathogen Biology, Chinese Academy of Medical Sciences & Peking Union Medical College
Address:	No. 9 Dong Dan San Tiao, Dong Cheng District, Beijing, P.R. China
Sample ID given by the sample provider:	IPBCAMS-WH-01
Submitting lab:	Institute of Pathogen Biology, Chinese Academy of Medical Sciences & Peking Union Medical College
Address:	No. 9 Dong Dan San Tiao, Dong Cheng District, Beijing, P.R. China
Sample ID given by the submitting laboratory:	BetaCoV/Wuhan/IPBCAMS-WH-01/2019
Authors:	Lili Ren, Jiarwei Wang, Qi Jin, Zichun Xiang, Zhiqiang Wu, Chao Wu, Yiwei Liu
Submitter information	
Submitter:	Lili Ren, Jiarwei Wang, Qi Jin, Zichun Xiang, Zhiqiang Wu, Chao Wu, Yiwei Liu
Submission Date:	2020-01-11
Address:	No. 9 Dong Dan San Tiao, Dong Cheng District, Beijing, P.R. China

Important note: In the GISAID EpiFlu™ Database Access Agreement, you have accepted certain terms and conditions for viewing and using data regarding influenza viruses. To the extent the Database contains data relating to non-influenza viruses, the viewing and use of these data is subject to the same terms and conditions, and by viewing or using such data you agree to be bound by the terms of the GISAID EpiFlu™ Database Access Agreement in respect of such data in the same manner as if they were data relating to influenza viruses.

>hCoV-19/Wuhan/IPBCAMS-WH-01/2019|EPI_ISL_402123|2019-12-24
 ATTAAGGTTTATACCTTCCAGGTAACAAACCAACCAACTTTCGATCTCTTGTAGATCTGTTCTCTAAACGAACTTAA
 AATCTGTGTGGCTGCACTCGGCTGCATGCTTAGTGCACTCACGCAGTATAATTAATAACTAATTACTGTCGTTGACAGG

TGAAC TTTCATTAATGACTTCTATTTGTGCTTTTTAGCCTTTCTGCTATTCCTTGTTTTAATTATGCTTATTTATCTTTT
GGTTCTCACTTGAAC TGCAAGATCATAATGAACTTGTACGCCTAAACGAACATGAAATTTCTTGTTTTCTTAGGAATC
ATCACAAC TG TAGCTGCATTTACCAAGAATGTAGTTTACAGTCACTCAACATCAACCATATGTAGTTGATGACCC
GTGTCCTATTCAC TTCTAATGATATTTAGAGTAGGAGCTAGAAAATCAGCACCTTAAATTGAATTGTGCGTGG
ATGAGGCTGGTTCTAAATCACCCATTAGTACATCGATATCGGTAATTATACAGTTTCCTGTTACCTTTTACAATTAAT
TGCCAGGAACCTAAATGGGTAGTCTTGTAGTGCCTTGTTCGTTCTATGAAGACTTTTTAGAGTATCATGACGTTCCGTGT
TGTTTTAGATTTCACTAAACGAACAACTAAATGTCTGATAATGGACCCAAAATCAGCGAAATGCACCCCGCATTAC
GTTTTGGTGGACCCTCAGATTCAACTGGCAGTAACCAGAATGGAGAACGCAGTGGGGCGGATCAAAAACAACGTGGGCCCC
AAGGTTTACCCAATAACTGCGTCTTGGTTCCACCGCTCTCACTCAACATGGCAAGGAAGACCTTAAATCCCTCGAGGA
CAAGGCGTTCCAATTAACACCAATAGCAGTCCAGATGACCAAATGGCTACTACCGAAGAGCTACCAGACGAATTCGTGG
TGGTGACGGTAAAATGAAAGATCTCAGTCCAAGATGGTATTTCTACTACTAGGAACTGGGCCAGAAGCTGGACTTCCCT
ATGGTGTAACAAAGACGGCATCATATGGGTTGCAACTGAGGGAGCCTTGAATACACCAAAAAGATCACATTGGCACCCGC
AATCTGCTAACAAATGCTGCAATCGTGTACAACCTTCTCAAGGAACAACATTGCCAAAAGGCTTCTACCGAGAAGGGAG
CAGAGGCGGCAGTCAAGCCTTCTCGTTCCTCATCAGTAGTCGCAACAGTTCAAGAAATCAACTCCAGGCAGCAGTA
GGGGAAC TTCTCCTGCTAGAATGGCTGGCAATGGCGGTGATGCTGCTCTTGCTTTGCTGCTGCTTGACAGATTGAACCAG
CTTGAGAGCAAAAATGCTGGTAAAGGCCAACAAACAAGGCCAAACTGTCACTAAGAAATCTGCTGCTGAGGCTTCTAA
GAAGCCTCGGCAAAAACGTA CTGCCACTAAAGCATAACAATGTAACACAAGCTTTCCGGCAGACGTGGTCCAGAACAACCC
AAGGAAATTTGGGGACCAGGA ACTAATCAGACAAGGA ACTGATTACAAACATTGGCCGCAAAATGCACAATTTGCCCCC
AGCGCTTACGCGTTCTTCGGAATGTCCGCAATTGGCATGGAAGTCACACCTTCGGGAACGTGGTTGACCTACACAGGTGC
CATCAAATGGATGACAAAGATCCAAATTTCAAAGATCAAGTCATTTTGCTGAATAAGCATATTGACGCATACAAAACAT
TCCCACCAACAGAGCCTAAAAGGACAAAAGAAGAAGGCTGATGAAACTCAAGCCTTACCGCAGAGACAGAAGAAACA
GCAAACTGTGACTCTTCTCCTGCTGCAGATTTGGATGATTTCTCCAAACAATTGCAACAATCCATGAGCAGTGTGACTC
AACTAGGCC TAAACTCATGCAGACCACACAAGGCAGATGGCTATATAAACGTTTTCGCTTTTCCGTTTACGATATATA
GTCTACTCTTGTGCAGAATGAATTTCTGTA ACTACATAGCACAAGTAGATGTAGTTAACTTTAATCTCACATAGCAATCT
TTAATCAGTGTGTAACATTAGGGAGGACTTGAAGAGCCACCACATTTTACCGAGGCCACGGGAGTACGATCGAGTGT
ACAGTGAACAATGTAGGGAGAGCTGCCTATATGGAAGAGCCCTAATGTGTA AAAATTAATTTTAGTAGTGTATCCCCAT
GTGATTTTAAATAGCTTCTTAGGAGAATGACAAAAA AAAAAAAAAAAAAAAAAAAAAAAAAA